# CLIN: A Continually Learning Language Agent for Rapid Task Adaptation and Generalization

**Bodhisattwa Prasad Majumder**[1], **Bhavana Dalvi Mishra**[1],
**Peter Jansen**[1,2], **Oyvind Tafjord**[1], **Niket Tandon**[1], **Li Zhang**[3],
**Chris-Callison Burch**[3], **Peter Clark**[1]
[1]Allen Institute of AI
[2]University of Arizona
[3]University of Pennsylvania

Contact: {bodhisattwam, bhavanad}@allenai.org
Project page: https://allenai.github.io/clin/

## Abstract

Language agents have shown some ability to interact with an external environment, e.g., a virtual world such as ScienceWorld, to perform complex tasks, e.g., growing a plant, without the startup costs of reinforcement learning. While recent work, e.g., Reflexion, has demonstrated how such agents can also self-improve by adding a textual memory of "hints" learned from prior experience, such improvements have been limited both in size and scope. In contrast, our goal is a language agent that can robustly improve performance over time, including when both the task and environment are varied. Our approach is to have the agent learn a textual representation of how the world works (rather than just isolated hints), expressed as a memory of *causal abstractions*, to guide future decision-making. In experiments, we find CLIN is able to continually improve on repeated trials on the same task and environment, outperforming state-of-the-art reflective language agents like Reflexion by 23 points in ScienceWorld and 1.4 points in ALFWorld benchmarks. CLIN can also transfer its learning to new environments and tasks, enhancing performance by 21 points in ScienceWorld and 11 points in ALFWorld. This suggests that language agents with a textual causal memory can play a significant role in interactive environments, including being able to rapidly improve over time.

## 1 Introduction

Large language models (LLMs) have been increasingly used to interact with external environments (e.g., simulated worlds) as goal-driven agents (Reed et al., 2022). However, it has been challenging for these language agents to efficiently learn from trial-and-error as traditional reinforcement learning methods require extensive training samples and expensive model fine-tuning (Chen et al., 2021; Ammanabrolu et al., 2020). More recently, new techniques have appeared in which an agent reflects on its own past experience solving a task in a particular environment, and generates language-based insights to help it retry the task, e.g., (Shinn et al., 2023; Zhao et al., 2023). Such methods have the advantage of not requiring parameter updates (particularly with frozen LLMs). However, the style of such insights plays a crucial role, and not all insights improve generalization performance. For example, a specific insight such as "I should go to desk 1 and find the lamp" (Shinn et al., 2023) may have limited value (or even hurt) for a different environment or task.

Our goal is a system that will continually improve over time, both while attempting the same task in the same environment, and across different tasks and environments. Our approach builds on prior work on reflection in two ways: First, we conjecture that a specific *style* of insight will be useful, namely one that captures **causal abstractions** about agent's actions,

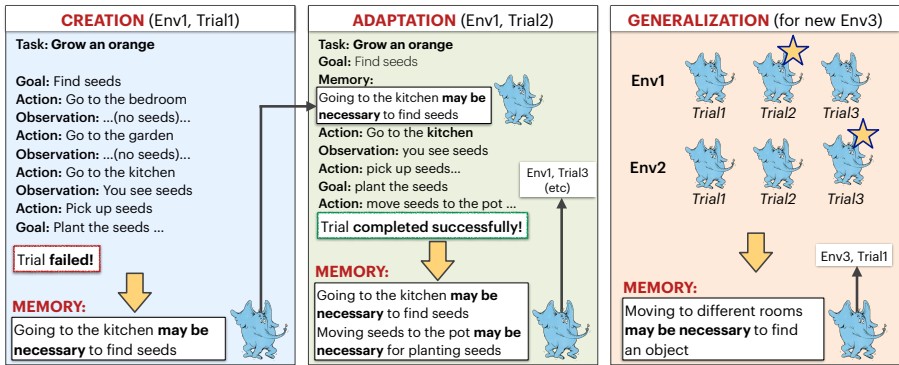

Fig. 1: CLIN creates (Trial1) or adapts (Trial2+) a **memory of causal abstractions** to help in future trials by reflecting on the last trial and current memory (Sec. 3.4 **Adaptation**). Here, reflecting on Trial1, CLIN notes in memory that going to the kitchen helped with finding seeds, enabling it to find the seeds faster in Trial2. To further generalize in new environments, CLIN generates a summary ("meta-memory") of the best (starred) memories from each prior episode, here generating the generalization that moving to different rooms helps finding objects (Sec. 3.4 **Generalization**).

e.g., "opening doors may be necessary for movement between rooms". Causal abstractions can potentially help the agent decide which action to take in the future, and can be viewed as a kind of action model learning (Arora et al., 2018), but placed in the modern context of language models. Second, we maintain these abstractions in a **continually evolving, dynamic memory**, which is regularly updated as the agent gains experience, allowing useful causal knowledge to persist (and unhelpful knowledge to be dropped) over time and between tasks and environments, as illustrated in Fig. 1.

We operationalize and evaluate this approach in a memory-augmented language agent called CLIN (**c**ontinual **l**earning from **in**teractions). CLIN is an agent that operates in a virtual, text-based environment (e.g., ScienceWorld (Wang et al., 2022), ALFWorld (Shridhar et al., 2021)) in which an agent is tasked with goals, e.g., boiling a liquid or growing a plant. We find that CLIN is able to rapidly learn about the environment and its action vocabulary and continually improve on repeated trials on the same task and environment, outperforming state-of-the-art (SOTA) reflective language agents like Reflexion by 23 points in ScienceWorld. CLIN can also transfer its learning to new environments (or tasks), through continual memory updates and achieving 21 (20 for new tasks) points performance boost. Similarly, in ALFWorld, CLIN enhances its base performance by 11 points in unseen tasks/environments. Our contributions are as follows:

- We describe and evaluate CLIN, a novel architecture for nonparametric agent learning. We show that using a dynamic, evolving memory over time, including over new tasks and environments, CLIN learns faster than the shorter-term memory approaches used in Reflexion and other memory-based agents, and with memories that generalizes better to new tasks and environments.
- We show that a memory of causal abstractions (or "action models") helps agents learn over an extended period. While action models have been used in formal planning before, we are the first to apply this concept to language agents.

Overall, this work suggests that a dynamic memory, centered around causal knowledge, is a promising way forward for agents built on frozen models to continually improve over time.

## 2 Related Work

**Reinforcement Learning.** There is a long literature of work on agents that can navigate complex environments. A common approach is to use reinforcement learning (RL), e.g., DRRN (He et al., 2015), KG-A2C (Ammanabrolu & Hausknecht, 2020), CALM (Yao et al., 2020), where agents learn a task over repeated trials. However, while effective, such agents typically require a large number of trials to learn and have trouble adapting to unexpected changes in the test environment. More recently, (Adaptive-Agent-Team et al.,

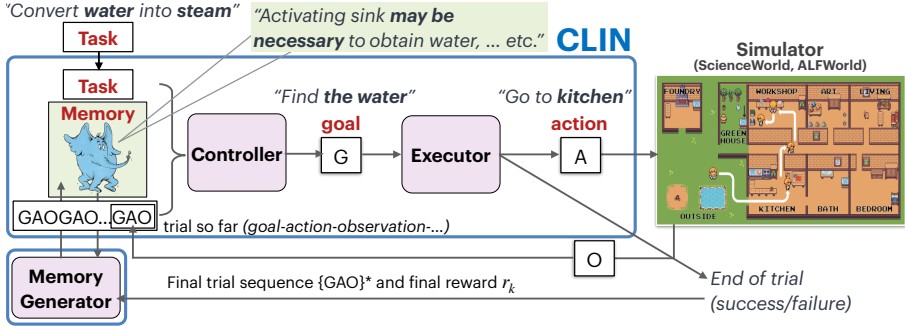

Fig. 2: In CLIN, a **controller** takes the current task, memory, and the trial so far, to generate the next goal. The **executor** converts the goal to a valid action. The simulator performs the action and returns an observation. Memory is updated at the end of each trial by the **memory generator** (Sec. 3.4).

2023) demonstrated AdA, an agent that could rapidly adapt to open-ended novel 3D problems, using meta-reinforcement learning, essentially being able to change its policy on the fly. However, AdA required vast amounts of pretraining, and was limited to the style of environments and problems seen in pretraining.

**Language Agents.** LLMs have provided a new tool for building goal-directed agents (Huang et al., 2022). Given a linguistic description of the world, the LLM can be prompted to suggest the next actions to achieve a goal requiring little training, e.g., SayCan (Ahn et al., 2022), ReAct (Yao et al., 2022), and SwiftSage (Lin et al., 2023), which combines a supervised agent and a deliberative agent together. However, while performing reasonably with little training data, such agents are unable to learn and adapt from experience.

**Reflection and Memory.** More recently, several language agents for interactive environments have been developed that can reflect on their own performance, and create a non-parameteric memory of language-based insights to help in new tasks via prompting (Brown et al., 2020). Voyager (Wang et al., 2023) operates in the world of Minecraft, growing a (code-based) skill library from rich feedback about the causes of its failures. ExpeL (Zhao et al., 2023) acquired "insights" by reflecting on traces of its behavior (trajectories) in a training phrase, for later use, but did not operate in a continuous fashion. (Chen et al., 2023) learned a memory helpful "tips" by reflecting on prior trajectories, including comparing (human) expert and system trajectories. Rememberer (Zhang et al., 2023) learned "encouraged" and "discouraged" actions from prior experience, but did not generalize explicit reasons for those preferences. Finally, Reflexion (Shinn et al., 2023) improved at a task by reflecting on a failed attempt, then learning specific reflections to help when retrying that same task (but not beyond). Our work is inspired by and goes beyond these, to create a system that learns in a continually evolving setting, including across changing environments and tasks, using a memory of causal rules about how the world works.

More generally, others have found that a memory of useful learnings can be used to improve frozen LLM behavior, e.g., in QA (Dalvi et al., 2022; Tandon et al., 2022; Madaan et al., 2023), or for modeling social behavior (Park et al., 2023). We apply this to goal-directed agents.

Finally, we note that the *content* of experiential memory is also important. Specifically, CLIN learns a memory of *causal abstractions*, which can be seen as learning a linguistic form of action model, describing the causal effects of actions. While there has been work in the planning community of learning action models in a formal context (Arora et al., 2018; Aineto et al., 2018), CLIN loosely applies this idea in the linguistic world of LLM agents.

# 3 Approach

## 3.1 Acting in the World

We follow the normal formalization for an agent performing actions in a partially observable environment, but add a memory $S$ as an additional input for decision making. The memory

contains learned task/environment knowledge to help the agent make better decisions in the next trial, and is updated at the end of each trial (described shortly). At each time step $t$, given a task $m$ (e.g., "grow an orange"), memory $S$, and the history of actions so far, the agent decides on its next goal $g$ and action $a$ in pursuit of that goal. In response, the environment returns the result of executing $a$ in the form of an observation $o$ and a reward $r$. This repeats until an end state is reached (such as completing, failing, or timing out). Thus at each step $t$, the history so far is $\mathcal{T}_{\leq t} = \{g_i, a_i, o_i\}^*_{i \leq t}$, and the agent's decision-making task at each time step $t$ can be described as:

$$m + e + \mathcal{T}_{\leq t} + S \rightarrow g_{t+1} \rightarrow a_{t+1} \tag{1}$$

We describe the implementation of this shortly. The full sequence of steps $\mathcal{T}_{end}$ to an end-state is called a *trial*. If the *same* task $m$ is attempted in the *same* environment $K$ times (resetting the environment each time), we call the collection of the $K$ trials an *episode*.

## 3.2 Continual Learning

The agent's goal is to maximize its reward $r$ (e.g., completing the task) on new trials. Learning occurs by updating the dynamic memory $S$ between (not during) trials, using a memory update function that takes memories from old trials as input, and generates a new memory for the next trial. Note that the memory does not grow monotonically; it may drop previous memory items and add new ones. Also note it is not perfect; some memory items may be erroneous, and ideally be dropped or modified in subsequent iterations. Learning is continuous in the sense that each new memory is generated from an ever-growing collection of previous memories. We define two classes of learning:

1. **Within-episode learning ("adaptation")** - same task $m$ and environment $e$. After each trial $k$, a new memory $S_{k+1}$ is generated from the most recent trial history $\mathcal{T}_k$ and final reward $r_k$, and memories from prior trials:

$$\mathcal{T}_k + r_k + \{S_{\leq k}\} \rightarrow S_{k+1} \tag{2}$$

   $S_{k+1}$ is then used to retry the same task $m$, next trial.
2. **Cross-episode learning ("generalization")** - new task $m_{new}$ or environment $e_{new}$. Given a *new* task or environment, an initial starting memory is generated using memories from *other* episodes. Specifically, we select the "best" (defined later) memory from each prior episode as inputs, and generate a new memory for use in this unexplored task/environment as output:

$$m_{new} + e_{new} + \{S_{best}, r_k\}_{prior\_episodes} \rightarrow S_{new} \tag{3}$$

Because this new memory generalizes prior memories, we also refer to it as a "meta-memory". Note that some generalizations in $S_{new}$ may be overly specific or wrong. However, we see both a net initial benefit in using $S_{new}$, and further task improvement in subsequent trials as $S_{new}$ is refined (adaptation), described later in Sec. 4.

## 3.3 CLIN: Agent Architecture

Our implementation, called CLIN, comprises three components for acting: the **memory**, a **controller**, and an **executor**. Learning then occurs using a fourth module, a **memory generator**, to generate an updated memory after each trial. These are illustrated in Fig. 2.

**Memory.** CLIN's memory ($\mathcal{S}$) is a persistent, dynamic collection of NL sentences expressing CLIN's current understanding of actions and their effects. Specifically, each sentence expresses a *causal abstraction* between actions, e.g., "opening the fridge is `necessary` to access apple juice", as well as negative learnings, e.g., "moving to another room `does not contribute` to freezing mercury.". Such statements are learned from past experiences (described shortly). Their role is to help CLIN make better action choices (Eqn. 1). Causal abstractions constitute CLIN's current understanding of the way the world behaves, and can be viewed as a modern version of *action models* used in formal planning, describing the effects of actions in the world (Arora et al., 2018).

**Algorithm 1** CLIN

1: **procedure** ADAPT(Task: $m$, Env: $e$, Memory: $\mathcal{S}$):
2:      Initialize Memory: $\mathcal{S}_0$
3:      **for** $k \in 1, \cdots, K$ **do**:
4:          Intialize Trial $\mathcal{T}$, $t$
5:          **while** $t <$ max. steps or task not complete **do**:
6:              $g_t = \texttt{Controller}\,(m, e, \mathcal{T}_{<t}, \mathcal{S}_{k-1})$
7:              $a_t = \texttt{Executor}\,(g_t, \text{admissible actions})$
8:              $r_t, o_t = \texttt{Simulator}\,(\mathcal{T}_{<t}, a_t)$
9:              $\mathcal{T}_{<t+1} = \mathcal{T}_{<t} + (g_t, a_t, o_t, r_t)$
10:         Final reward $r_k = r_t$
11:         $\mathcal{S}_k = \texttt{memory-generator}\,(\{\mathcal{S}_{<k}\}, \mathcal{T}_k, r_k)$
12: **end procedure**
13: **procedure** GENERALIZE(Task: $m$, Env: $e$, past $m'/e'$)
14:      $\{\mathcal{S}_{\text{crucial}}, r_k\} = \texttt{crucial-memories}\,(\text{past } m'/e')$
15:      $\mathcal{S}_{\text{meta}} = \texttt{meta-memory}\,(\{\mathcal{S}_{\text{crucial}}, r_k\}, m)$
16:      ADAPTATION($m$, $e$, $\mathcal{S}_{\text{meta}}$)
17: **end procedure**

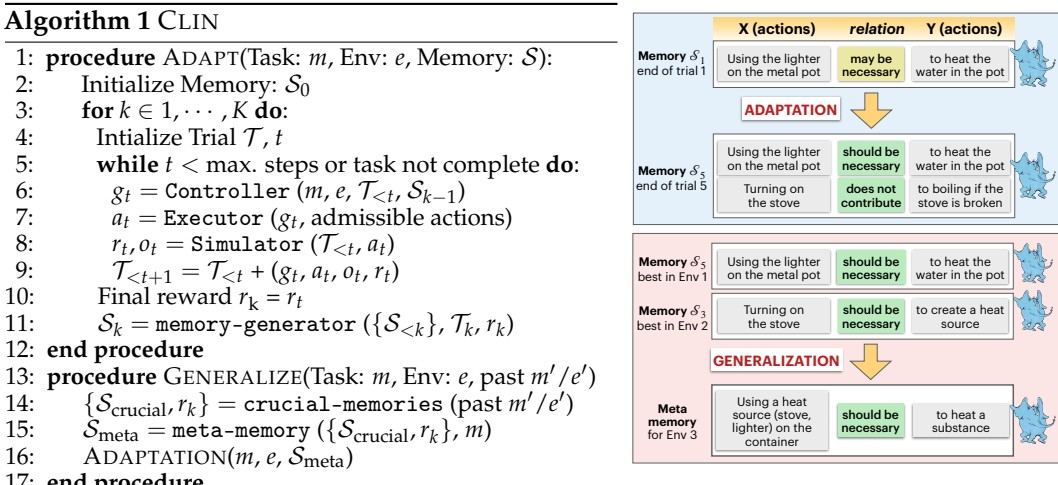

Fig. 3: (LEFT) CLIN's continual learning algorithm. (RIGHT) Example causal abstractions.

**Controller.** At each time step in a trial, the controller generates the next goal to pursue in service of the overall task $m$. In CLIN the controller is a frozen LLM, whose prompt includes the current **task** $m$, e.g., "convert water into steam", selected statements from the **current memory** $\mathcal{S}$ (a list of sentences), and the **trial so far** (the sequence of goal-action-observation triples. It is prompted to output the next **goal** $g_{t+1}$ to pursue, e.g., "find water". Note we only use selected statements from memory (rather than the whole memory), in order to avoid irrelevant knowledge distracting the controller. Selection is itself done with a separate query to the LLM, prompting it to list relevant memory items, with the full memory $S$ and the task included in that prompt.

**Executor.** The role of the executor is to convert the generated goal $g_{t+1}$ into a valid **action** $a_{t+1}$ that can be executed in the environment in pursuit of that goal. In other words, it serves to map goals into the specific action space of the environment. Again a (frozen) LLM is used, whose prompt includes the goal $g_{t+1}$ (from the controller, above), the trial so far, and all the possible actions that can be performed in the current state (provided by the environment, as is standard practice in current generative agent research (Ahn et al., 2022; Yao et al., 2022; Lin et al., 2023; Park et al., 2023)). The list of possible actions is expressed as possible action templates and available objects that can instantiate them, rather than a combinatorially large enumeration of possible actions. The model is then prompted to generate a candidate action to perform (see prompt in Fig. 6). Finally, CLIN checks this candidate action is one of the valid actions. If it is not, it finds the most similar valid action using the pre-trained embeddings from the sentence-transformer model (Reimers & Gurevych, 2019). If the top-ranked valid action has a similarity score greater than a threshold (here, 0.9, chosen as a hyperparameter), the action is selected. Otherwise, we perform iterative refinement (Madaan et al., 2023) by suffixing the context with feedback stating the candidate action is not executable. This allows the executor to retry generation up to a max number (here, 5).

Finally, upon executing the action $a_{t+1}$, CLIN receives a partial next state, as an **observation**, from the environment and the reward ($r$) $\in [0, 1]$, provided by the environment. A snapshot of a full trial is given in lines 4-10 in Alg. 1.

Note that CLIN does not make use of any gold data to identify goals and memories. Rather, we expect CLIN to perform a balanced act of exploration-exploitation by interacting, learning, and adapting to unseen tasks or environments—a key difference from few-shot generative agents (Ahn et al., 2022; Yao et al., 2022; Lin et al., 2023; Park et al., 2023).

### 3.4 Continual Learning in CLIN

At the end of each trial (completion or failure), CLIN uses a **memory generator** to create or update its memory. The memory generator is a (frozen) LLM prompted to reflect on the current trial and memory, and generate a new memory of insights in the form of (English sentences expressing) useful **causal abstractions**.

To make the LLM generate causal abstractions, we use special instructions in the prompt that ask the LLM to generate insights in a particular templated syntax (see prompt in Fig. 7). To capture actions enabling desired changes and helpful state transitions, we use the template "X is NECESSARY to Y", and to capture contrastive examples of unsuitable actions and state transitions, we employ "X DOES NOT CONTRIBUTE to Y" (Fig. 3), where X, Y are related to actions. These abstractions are functionally analogous to hindsight experience replay (Andrychowicz et al., 2017), obtained from CLIN's past self-explorations. To allow the LLM to express uncertainty, we encourage it to use modifiers: "X may ..." for moderate to high uncertainty, and "X should ..." for low uncertainty (See Fig. 3).

As described earlier, there are two kinds of memory update needed: (a) re-generating the memory when retrying the same task in the same environment ("adaptation") (b) re-generating the memory for a *new* task/environment ("generalization"), as we now describe.

**Within-episode learning ("adaptation").** To update the memory after each trial within an episode (Eqn. 2), the memory generator is prompted with the most recent trial (a sequence of $(g_t, a_t, o_t)$ tuples and the final reward $r_k$[1] ), and the memories from the three most recent trials $\{\mathcal{S}_{k-2}, \mathcal{S}_{k-1}, \mathcal{S}_k\}$. It is then prompted to generate an updated memory $\mathcal{S}_{k+1}$, namely a new list of semi-structured causal abstractions in the forms described above, for use in the next trial. Although we do not specify a maximum size for the memory, we observe that size of the generated memory (i.e., the number of causal abstractions generated) is far less than the number of actions executed in the trial, indicating the memory-generator additionally performs a saliency-based pruning to keep only important insights based on the success of the trial (final reward $r_k$ for trial $\mathcal{T}_k$).

**Cross-episode learning ("generalization").** Given a new task $m_{\text{new}}$ or environment $e_{\text{new}}$, the memory generator is prompted to generate a suitable memory, generalizing from the best memories from previous tasks/environments and suitable for this new situation (Eqn. 3) - a form of meta-learning (Hospedales et al., 2020). Following the prioritized level replay scheme (Jiang et al., 2021), we choose the most successful trial per episode (based on the reward $r_k$) and retrieve memories abstracted from those trials with a fixed archive of size 10, a hyperparameter. If the *environment* is new, the prompt instructs the LLM to generate a memory helpful "to solve the same task in a new environment configuration", given the new task description. The prompt is designed to encourage the LLM to generate generic causal insights about the task, not tied to specific environmental details (Fig. 8). Similarly, if the task is new, the prompt is modified accordingly (Fig. 9).

## 4 Experimental Setup

Test-time adaptation and generalization via continual learning require a variety of complex tasks and environment configurations to allow an agent to explore, learn latent causal insights from interactions, and exploit them in the future. We evaluate CLIN's performance in two benchmarks: **ScienceWorld** (Wang et al., 2022) and **ALFWorld** (Shridhar et al., 2021). Both benchmarks consist of a text-based interactive environment requiring complex interactive reasoning processes to solve a plethora of tasks. ScienceWorld focuses on science-based tasks[2] spanning 18 task classes (e.g., thermodynamics, genetics, friction, etc.) with several variations (total 164). ALFWorld has 6 categories of tasks: Pick, Clean, Heat, Cool, Look, and Pick-two-items, with a total of 134 environments.

---

[1]The reward is converted to NL feedback using 7 simple rules, e.g., "if score >= 0 and score < 20 then feedback = *The agent performed poorly and made some progress but not enough to solve the task.*"

[2]ScienceWorld tasks are grouped into Short (S), e.g., *pick & place* and Long (L), e.g., *grow plant*, based on the # of gold actions.

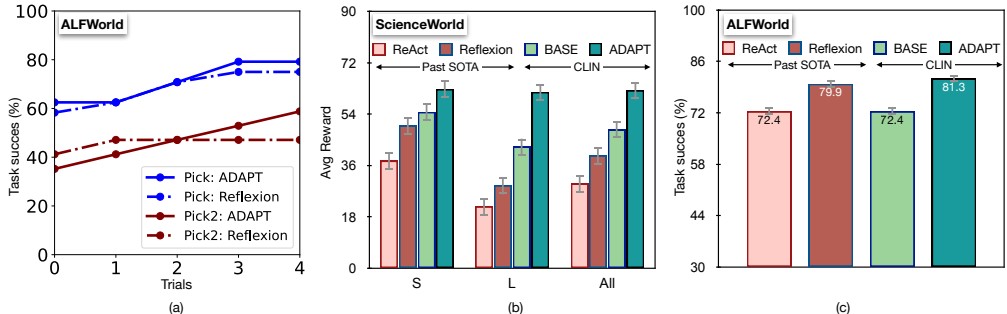

Fig. 4: **Rapid task adaptation with CLIN. (a)** Example tasks with CLIN's adaptation. For CLIN, Trial-0 is BASE, Trial-4 is ADAPT. Comparison of CLIN with Reflexion (Shinn et al., 2023) in **(b)** ScienceWorld and **(c)** ALFWorld. (More in Appendix C).

ALFWorld is used in several previous approaches (Yao et al., 2022; Shinn et al., 2023). However, most tasks in ALFWorld required a small number of actions and limited atomic skills e.g., pick, place, clean, etc. Our aim is show continual improvement in long and complex tasks. ScienceWorld entails tasks that require a median 37 steps, and each task can be tested in a variety of different environments, each with its unique objects/affordances.

**CLIN-BASE.** The standard base agent in previous work uses a few-shot ReAct prompt (Yao et al., 2022), which also applies to Reflexion. However, both in ScienceWorld and ALFWorld, gold trajectories from similar tasks overlap more than 80% with the target task trajectory. To avoid reliance on these demonstration examples, in CLIN, we use a zero-shot controller prompt (see in Fig. 6) as CLIN-BASE. We observe CLIN-BASE, powered by GPT-4 (Achiam et al., 2023), is superior on ScienceWorld (Fig. Figure 4b) and similarly performant on AlfWorld (Fig. 4c) than the one-shot ReAct agent. In our experiments, we use CLIN-BASE as the base agent for within- and cross-episode learning.

## 5  Results and Analysis

**A. How well does CLIN perform** *within-episode* **learning ("adaptation")?**

We refer to this setup as **ADAPT**. Here, we focus on CLIN's ability to adapt to a task by attempting it for several trials in the same environment configuration. Most importantly, CLIN initializes with an empty memory at the beginning of the first trial and generates memory at the end of each trial. While the environment gets reset at the trial boundary, CLIN's memory continues to be updated, capturing informative causal abstractions pertaining to both successful and failed actions. We compare with Reflexion (Shinn et al., 2023), a representative SOTA memory-based system that also reflects and learns from experience.

Results are shown in Fig. 4b (ScienceWorld) and Fig. 4c (ALFWorld). In both domains, after the trials, we see CLIN-ADAPT **improves over time** (+13.6 pts (all), compared to its starting performance BASE), and also that **the improvement is greater than that for Reflexion** (+9.8 pts compared to Reflexion's base performance, namely the non-memory ReAct). Fig. 4a also shows the learning curves for two tasks in ALFWorld. Again, we see both Reflexion and CLIN-ADAPT improve over time, but again with greater improvement for CLIN-ADAPT (+16.7% for Pick, +23.6% for Pick2) than Reflexion (+16.7%, +5.9% respectively).

**B. How well does CLIN generalize for a fixed task, across** *different* **environments?**

In this setup, called (**GEN-ENV**), for a task $m$, we run CLIN for 10 different (train) environment settings (with varying objects and starting locations) and then create meta-memories from its exploration to solve the same task in an unseen (test) environment. If CLIN, in GEN-ENV does not successfully complete the task in the new environment, it can continue learning and retrying that task. We refer to this setup as GEN-ADAPT(G+A).

Here, we compare CLIN with RL methods DRRN (He et al., 2015), KG-A2C (Ammanabrolu & Hausknecht, 2020), and CALM (Yao et al., 2020) trained on all (large) training variations

| Type | RL Methods | | | Generative Language Agents | | | CLIN (ours) | | |
|------|------|-------|------|--------|-------|-----------|------|---------|------|
| | DRRN | KGA2C | CALM | SayCan | ReAct | Reflexion | BASE | GEN-ENV | G+A |
| **S** | 22.1 | 19.1 | 5.2 | 36.5 | 37.6 | 49.9 | 54.7 | 58.3 | **71.0** |
| **L** | 11.2 | 6.2 | 1.9 | 29.2 | 21.5 | 28.9 | 42.5 | 47.1 | **68.0** |
| **All** | 16.7 | 12.7 | 3.6 | 32.9 | 29.6 | 39.4 | 48.6 | 52.7 | **69.5** |

Table 1: **Generalization** across unseen environments in ScienceWorld. Task-wise results in Table 6.

| Method | Pick | Clean | Heat | Cool | Look | Pick2 | All |
|--------|------|-------|------|------|------|-------|-----|
| ReAct | 58.3 | 71.0 | 87.0 | 81.0 | 94.4 | 41.2 | 72.4 |
| Reflexion | 75.0 | 74.2 | **91.3** | 90.5 | **100.0** | 47.1 | 79.9 |
| CLIN (ours) | | | | | | | |
| GEN-ENV+A | **83.3** | **77.4** | 87.0 | **95.2** | **100.0** | 58.8 | **83.6** |
| GEN-TASK+A | 79.2 | 74.2 | **91.3** | 90.5 | **100.0** | **64.7** | 82.8 |

Table 2: **Generalization** across unseen environments and tasks in ALFWorld

with simulator reward and Generative Language agents, SayCan (Ahn et al., 2022), ReAct (Yao et al., 2022), and Reflexion (Shinn et al., 2023), prompted with few-shot demonstrations.

Table 1 compares CLIN with baselines that learn from training environmental variants for a task to improve its performance in a novel environment [3]. Language agents (including CLIN) that use NL feedback from the ScienceWorld (e.g., "Door to the kitchen is closed") perform significantly better compared to RL methods that purely rely on (sparse) numeric rewards from the environment to learn a policy.

We also observe that **CLIN's meta-memory is overall advantageous**, helping CLIN with a stronger start on unseen environments (52.7, GEN-ENV all) compared with no memory (BASE, 48.6). Note that advantage is not guaranteed, as for some tasks, the memories may be too specific and hurt, but there is an overall net benefit. **This gain persists** through continued learning in these new environments, in which the meta-memory is refined, resulting in an overall net gain of 20.9% (all) over base performance, **substantially higher than for Reflexion** (+9.8% (all) compared with its no-learning version ReAct). Fig. 5a plots this improvement over time and also compares it to improvement without a meta-memory, showing persistent net improvement.

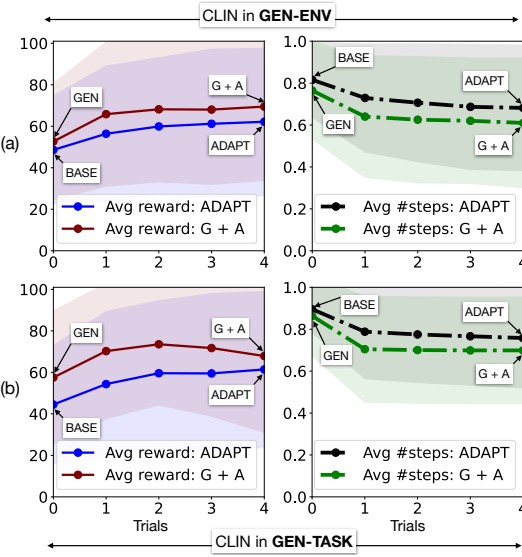

Fig. 5: Reward and #steps trends in **(a)** GEN-ENV and **(b)** GEN-TASK for ScienceWorld.

Similarly, for the ALFWorld, in the GEN-ENV setting, CLIN uses memories learned from earlier (different) environments. Table 2 shows that CLIN **generalizes better than Reflexion to new environments**, namely from its base performance (same as ReAct) to 83.6% (+11.2%), vs. +7.5% (Reflexion). The gains are likely smaller here because ALFworld is substantially simpler than ScienceWorld.

**C. How well does CLIN generalize across *different* tasks, in a fixed environment?**

This setup is referred to as **GEN-TASK**. For an environment $e$, we run CLIN to solve a task $m$ and then condense its learning to solve a novel task $m'$ in the environment $e$. If CLIN does not successfully complete the new task, it can continue learning and retrying that task. We refer to this setup as GEN-ADAPT(G+A). We took all test examples where we have a different task defined in the same environment configuration. (Adaptive-Agent-Team et al., 2023) suggests that transferring learning from a random task can be very hard; hence we

---

[3]Baseline numbers are derived from Table 1 in (Lin et al., 2023)

couple tasks that are related (revolve around overlapping task-critical objects/locations such water, kitchen), such as *boil* and *freeze* to measure transfer learning from one to the other. This is a novel setup where we do not have any off-the-shelf baselines. However, here, we compare against CLIN-BASE, a strong baseline.

Mirroring trends from GEN-ENV, CLIN demonstrates **strong transfer learning to new tasks** (Fig. 5b) with +13% improvement over its BASE performance, being better at 38.8% of datapoints. The improvement comes from critical learnings about the environment (e.g., "apple juice is in the fridge", required for both boiling and freezing it), leading to improvement in previously low-performing tasks in both ADAPT and GEN-ENV setups.

For the ALFWorld benchmark, in the GEN-TASK setting, CLIN had access to memories from other tasks of same type. Fig. 2 shows that CLIN can generalize its learnings across tasks to improve its success rate by 10.4% from its base performance (here same as ReAct).

**D. Does CLIN become more efficient during generalization?**

**Yes.** Fig. 5(a),(b) shows that the transfer learning in GEN-ENV, GEN-TASK and G+A helps CLIN to solve the tasks with fewer steps[4] and achieve higher rewards.

**E. Does a memory of causal abstractions help more than free-form advice ("hints")?**

**Yes.** CLIN extracts causal abstractions structured around 'necessary' and 'does not contribute' relations. To ablate this, we removed this constraint on generation and instead prompted CLIN to generate free-form advice for future trials. We observed CLIN then no longer generated causal knowledge but instead generated helpful hints (see Fig. 13). In ScienceWorld, the average reward drops by 6 pts (in 10% cases than CLIN), and in ALFWorld, the success rate drops by 1.4% when using the unstructured memory, indicating the usefulness of causal abstractions, as shown in Table 3.

| CLIN in | $\Delta$avg score ($\downarrow$) | %ep. drop. ($\uparrow$) |
|---|---|---|
| ScienceWorld | 6.2 | 10.0 |
| ALFWorld | – | 1.4 |

Table 3: Ablation for causal memory

**F. Do causal abstractions capture task- or environment-specific information?**

Causal abstractions in CLIN capture all necessary information about the world required as needed—1) **environment**: "Searching on sofa should be necessary to find the second keychain", 2) **task**: "picking the keychain should be necessary to put it on the table", and 3) **explorative sub-goals** that are not achieved yet: "Checking other locations like drawers and shelves may be necessary to finding the second CD."

**G. How does CLIN recover from unhelpful memories?**

While the final performance with memory is indicative of their effectiveness, we performed additional human evaluation of generated memory insights to evaluate their correctness across trials. For generalization setups, we ran-

| Insights | ScienceWorld | | | | ALFWorld | |
|---|---|---|---|---|---|---|
| | GEN-ENV | G+A | GEN-TASK | G+A | GEN | G+A |
| Total | 100 | 105 | 98 | 107 | 94 | 92 |
| Correct | 72.0% | **91.4%** | 73.9% | **91.1%** | 72.3% | **90.1%** |

Table 4: Memory correctness for CLIN

domly 10 task-environment combinations to evaluate the correctness of memories used in them, notably the meta-memory used for trial 0 (GEN) and memory adapted for the best trial (GEN-ADAPT). Two annotators rated the insights (cohen's $\kappa = 0.78$) for correctness with reference to gold trajectories. Table 4 shows that some meta-memories may not be applicable initially; however, with adaptation, in later trials, the correctness of the memory insights significantly improves, leading to a direct increase in task performance. However, CLIN is unlikely to generate an insight related to an unobserved activity, which could result in less useful insights and lower future performance in some cases. A curiosity-driven agent could mitigate this issue (Burda et al., 2019), which we leave as future work.

---

[4]# steps in Fig. 5(a),(b) are normalized between 0-1, 1 being maximum #steps allowed for a task.

## 6  Conclusion

Our goal is a system that can continually improve over time, both while rapidly adapting to a task by multiple retries and efficiently generalizing to novel tasks and environments. We propose CLIN, an architecture for language agents that constructs a persistent, dynamic memory of causal abstractions, refines it over time, and uses it to improve its performance on future tasks, achieving state-of-the-art performance. Our evaluation suggests that this architecture is effective, and we hope can thus help future development of language agents.

**Acknowledgement**   We sincerely thank Aristo team members Tushar Khot, Ashish Sabharwal, Shashank Gupta, Nathaniel Weir, Kyle Richardson, Jiangjie Chen, Archiki Prasad, and other members such as Faeze Brahman, Alexander Koller at the Allen Institute for AI for their generous feedback.

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

## A  Appendix

## A  CLIN prompts

Figures 6 to 9 are the complete prompts for next-action generation (controller + executor), memory-generator during ADAPT, GEN-ENV, and GEN-TASK.

## B  Example Memories

Example generated memory for ADAPT, GEN-ENV, and GEN-TASKsetups in Figures 10 to 12.

## C  More results

Full results for CLIN outperforming Reflexion is in Table 5. For the ScienceWorld benchmark, we exclude electricity tasks since they deviate from standard electrical conventions, prohibiting us from fairly using LLM agents. We choose the first 10 test variants for each 18 tasks selected. The full list of 18 tasks from the benchmark, with the number of test variants used in parentheses:

grow-plant (10), identify-life-stages-1 (5), grow-fruit (10), measure-melting-point-known-substance (10), mendelian-genetics-unknown-plant (10), chemistry-mix-paint-secondary-color (9), freeze (9), lifespan-longest-lived (10), inclined-plane-determine-angle (10), boil (9), use-thermometer (10), chemistry-mix (8), lifespan-shortest-lived (10), find-plant (10), find-living-thing (10), identify-life-stages-2 (4), mendelian-genetics-known-plant (10), inclined-plane-friction-named-surfaces (10).

Short tasks have oracle lengths less than 37 steps (median), and Long tasks have oracle lengths more than equal to 37 steps.

The map to the short names used for tasks in the paper:

Temp: use-thermometer, measure-melting-point-known-substance; Pick&Place: find-plant, find-living-thing; Chemistry: chemistry-mix, chemistry-mix-paint-secondary-color; Lifespan: lifespan-longest-lived, lifespan-shortest-lived; Biology: identify-life-stages-1, identify-life-stages-2, Boil; Freeze; Grow Plant, Grow Fruit; Force: inclined-plane-determine-angle; Friction: inclined-plane-friction-named-surfaces; Genetics: mendelian-genetics-known-plant, mendelian-genetics-unknown-plant.

```
[System]: You are an AI agent helping execute a science experiment in a simulated
environment with limited number of objects and actions available at each step.

[User]:
Possible objects ( value an OBJ can take ):
{objects_str}

Your next action should be in one of the following formats:
Possible actions:
{actions_str}

If I say \"Ambiguous request\", your action might mean multiple things. In that
case, respond with the number corresponding to the action you want to take.

What action would you like to do next?

First, scan the (unordered) list of learnings, if provided. Decide if any of the
learnings are applicable given the last observation to make progress in this task.
Then only use selected learnings, if any, to construct a rationale for picking the
next action. If no Learning is selected, construct the rationale based on the last
observation. Format your response as follows:

Write 'I used learning id(s):' as a comma separated list; the list can be empty if
no learnings selected. Then, write $$$ followed by the rationale. Finally, write
### followed by the single next action you would like to take.

If you think you have completed the task, please write TASK_COMPLETE as the next
action.

If the task requires you to 'focus' on something (OBJ), please write FOCUS ON <OBJ>
as the next action. FOCUS is a extremely critical action that can be only used the
number of times 'focus' is mentioned in the task description. Using it more than
that or inappropiately (such as on a wrong object) will terminate the session and
the task will be rendered as incomplete.

If you performed an action that requires waiting to see the effect, please write
'wait' as the next action.
```

Fig. 6: Prompt for the Controller and the Executor

**Superior BASE performance.** Figure 4 depicts a superior BASE performance for CLIN than the final performance of both ReAct and Reflexion despite using the same underlying

```
[System]: You are an expert assistant.

[User]:
You are given CURRENT TRACE, a sequence of actions that an agent made in a world
to accomplish a task.

Task is detailed at the beginning.
For each action, there is a rationale why the agent made that action.
There is an observation that provide details about the new state of the world
after each action was executed.
The CURRENT TRACE is accompanied by an EVALUATION REPORT indicating the success of
the attempt to the task.

You can also be provided with PREVIOUS LEARNINGS which are learnings from the
previous attempts by the agent for the same task in the same environment/world.
TASK indicates the task description. EPISODE indicates the number of previous
attempts of the task.

Generate a summary of learning, as a numbered list, that will help the agent to
successfully accomplish the SAME task AGAIN, in the SAME world.

Each numbered item in the summary can ONLY be of the form:
X MAY BE NECCESSARY to Y.
X SHOULD BE NECCESSARY to Y.
X MAY BE CONTRIBUTE to Y.
X DOES NOT CONTRIBUTE to Y.

{CURRENT TRACE}
Action: ...
Observation: ...
...
EVALUATION REPORT:
REWARD_FINAL: 100. This means: The agent has performed exceptionally well and
successfully solved the task.

Summary of learning as a numbered list:
```

Fig. 7: Prompt for CLIN's memory generator during ADAPT

```
[System]: You are an expert assistant.

[User]: You are given a collection of learning lists, that are derived from
actions made by an agent and subsequent observations from a world to accomplish a
TYPE of TASKs. All of these TASKs belong to a same TYPE (such as 'boiling') but
they are executed in different ENVIRONMENT configurations. A different ENVIRONMENT
configuration means there are presence of a different set of objects (lighter
instead of a stove) that are critical for solving the TASK, presence of a
different set of distractor objects that are not useful for the TASK, a different
floor plan, etc.

For each learning list, the TASK description is provided at the beginning as TASK:

Each learning list indicates a list of learnings from the agent's best attempt to
solve the TASK.

Each learning list is associated with an EVALUATION REPORT indicated how sucessful
the respective attempt was for solving the task.

Consider all learning lists and combine them in to a summary of learnings, as a
numbered list, that will help the agent to successfully accomplish a NEW TASK
related to the previous TASKs (such as 'boiing') in an ENVIRONMENT configuration
that it has not seen before. The NEW TASK description will be provided.

Each numbered item in the summary can ONLY be of the form:
X MAY BE NECCESSARY to Y.
X SHOULD BE NECCESSARY to Y.
X MAY NOT CONTRIBUTE to Y.
X DOES NOT CONTRIBUTE to Y.

{PREVIOUS LEARNINGS}
TASK: ...
LEARNINGS:...
EVALUATION REPORT:
REWARD_FINAL: 100. This means: The agent has performed exceptionally well and
successfully solved the task.
...

NEW TASK: ...
Summary of learning as a numbered list:
```

Fig. 8: Prompt for CLIN's memory generator during GEN-ENV

```
[System]: You are an expert assistant.

[User]: You may be given a list of learnings, that are derived from actions made
by an agent and subsequent observations from a world to accomplish a TASK in an
ENVIRONMENT CONFIGURATION.

For the learning list, the TASK description is provided at the beginning as TASK:

The learnings are from the agent's best attempt to solve the TASK.

The learning list is associated with an EVALUATION REPORT indicated how sucessful
the attempt was for solving the task.

Now, generate a summary of learnings from the existing ones if provided, such that
they will be useful to the NEW TASK in the SAME ENVIRONMENT CONFIGURATION. The NEW
TASK may require different actions which are not captured in the given learnings
but given learnings can be used to infer about the ENVIRONMENT CONFIGURATION. The
NEW TASK description will be given. If PREVIOUS LEARNINGS says 'No learnings
available', improvise learnings for the NEW TASK.

Each numbered item in the summary can ONLY be of the form:
X MAY BE NECCESSARY to Y.
X SHOULD BE NECCESSARY to Y.
X MAY NOT CONTRIBUTE to Y.
X DOES NOT CONTRIBUTE to Y.

{PREVIOUS LEARNINGS}
TASK: ...
LEARNINGS:...
EVALUATION REPORT:
REWARD_FINAL: 100. This means: The agent has performed exceptionally well and
successfully solved the task.
...

NEW TASK: ...
Summary of learning as a numbered list:
```

Fig. 9: Prompt for CLIN's memory generator during GEN-TASK

Task: Your task is to determine whether round seed shape is a dominant or recessive trait in the pea plant. If the trait is dominant, focus on the blue box. If the trait is recessive, focus on the orange box.

Memory (ADAPT), after Trial5:
1. Looking around MAY BE NECESSARY to gather information about the environment.
2. Moving to the greenhouse SHOULD BE NECESSARY to find the pea seeds.
3. Picking the jar SHOULD BE NECESSARY to access the pea seeds.
4. Moving the round green pea seed to flower pot 2 SHOULD BE NECESSARY to grow the pea plants.
5. Moving the wrinkly green pea seed to flower pot 3 SHOULD BE NECESSARY to grow the pea plants.
6. Waiting SHOULD BE NECESSARY for the pea plants to grow and show their traits.
7. Looking at the adult pea plant in flower pots 2 and 3 MAY BE NECESSARY to observe the traits of the offspring.
8. Focusing on the blue box SHOULD BE NECESSARY to indicate the trait of the seed shape.
9. Continuously looking at the same environment DOES NOT CONTRIBUTE to the task of determining the dominant or recessive trait in the pea plant.
10. Moving to the workshop DOES NOT CONTRIBUTE to determining the dominant or recessive trait in the pea plant.
11. Picking up the battery or the wires in the workshop DOES NOT CONTRIBUTE to finding the pea seeds.
12. Waiting for longer periods MAY BE NECESSARY to allow the pea plants to fully grow and show their traits.

Fig. 10: Memory generated after Trial5 in ADAPT for a Genetics task in ScienceWorld.

```
Task: Your task is to determine which of the two inclined planes (aluminum,
platinum) has the most friction. After completing your experiment, focus on the
inclined plane with the most friction.

Meta-memory (GEN-ENV):
1. Moving to the hallway SHOULD BE NECESSARY to reach the workshop.
2. Moving to the workshop SHOULD BE NECESSARY to find the block.
3. Picking up the block SHOULD BE NECESSARY to move it to the inclined planes.
4. Placing the block on the first inclined plane (either aluminum or platinum)
SHOULD BE NECESSARY to measure the friction.
5. Activating the stopwatch SHOULD BE NECESSARY to time the experiment.
6. Waiting for a certain period MAY CONTRIBUTE to observing the friction effect.
7. Deactivating the stopwatch SHOULD BE NECESSARY to stop timing the experiment.
8. Moving the block to the second inclined plane (either aluminum or platinum)
SHOULD BE NECESSARY to compare the friction.
9. Activating the stopwatch again SHOULD BE NECESSARY to time the second part of
the experiment.
10. Waiting for a certain period again MAY BE NECESSARY to observe the friction
effect.
11. Deactivating the stopwatch again SHOULD BE NECESSARY to stop timing the
experiment.
12. Focusing on the inclined plane with the most friction SHOULD BE NECESSARY to
conclude the experiment.
13. Repeating the experiment multiple times MAY BE NECESSARY for more accurate
results.
14. Looking around in the initial room multiple times DOES NOT CONTRIBUTE to the
task.
15. Moving the block back and forth between the two inclined planes DOES NOT
CONTRIBUTE to the task.
```

Fig. 11: Meta-memory used in GEN-ENV for a Friction task in ScienceWorld.

```
Task: Your task is to freeze mercury. First, focus on the substance. Then, take
actions that will cause it to change its state of matter.

Meta-memory (GEN-TASK):
1. Looking around MAY BE NECESSARY to identify the available resources and the
layout of the environment.
2. Moving to different rooms SHOULD BE NECESSARY to find the tools and materials
needed to change the state of the substance.
3. Picking up items like glass cups or metal pots SHOULD BE NECESSARY to contain
the substance for changing its state.
4. Focusing on the substance SHOULD BE NECESSARY to understand its properties and
how to interact with it.
5. Picking up the thermometer SHOULD BE NECESSARY to monitor the temperature of
the substance.
6. Using the thermometer on the substance SHOULD BE NECESSARY to monitor the
progress of the task.
7. Puring the substance into the container SHOULD BE NECESSARY to prepare it for
cooling.
8. Moving the container to a cooling device SHOULD BE NECESSARY to cool the
substance.
9. Waiting for a period of time after cooling the substance SHOULD BE NECESSARY to
allow the substance to change state.
10. Repeatedly checking the temperature of the substance SHOULD BE NECESSARY to
monitor the progress of the task.
11. Activating the stove DOES NOT CONTRIBUTE to the task as it does not progress
the task.
12. Picking up unrelated items like a lighter DOES NOT CONTRIBUTE to the task as
it does not progress the task.
13. Moving to unrelated rooms like the workshop DOES NOT CONTRIBUTE to the task as
it does not progress the task.
14. Teleporting to the kitchen MAY BE NECESSARY for the task as it speeds up the
process of moving between rooms.
15. Using the thermometer multiple times on the substance after it reaches
freezing point DOES NOT CONTRIBUTE to the task as it does not progress the task.
```

Fig. 12: Meta-memory used in GEN-TASK for a Freeze task in ScienceWorld.

```
Task: Your task is to make a red paper.

Ablated memory (GEN-TASK):
1. Avoid repeating actions: The agent performed the "look around" action twice at
the beginning of the task. This is unnecessary and wastes time. Once you have a
clear understanding of your surroundings, move on to the next step.
2. Be mindful of the order of actions: The agent moved to the workshop and read
the instructions after looking around twice. It's better to first read the
instructions to understand the task requirements.
3. Use the correct containers for mixing: The agent used a cup for mixing the
ingredients, which was successful. Continue using suitable containers like a cup
or a bowl for mixing.
4. Be clear with your actions: The agent made an ambiguous request when trying to
mix the cup containing red paint. Be specific with your actions to avoid
confusion.
5. Avoid unnecessary movements: The agent teleported to the art studio after
picking up the paper. Plan your actions carefully to avoid unnecessary movements.
6. Use the correct method to combine ingredients: The agent successfully mixed the
ingredients in the cup. Continue using suitable methods to combine the ingredients,
such as using a blender or a food processor.
7. Keep track of your actions: The agent moved the paper to the cup and then back
to the inventory. This is unnecessary and wastes time. Keep track of your actions
to avoid repeating them.
8. Always refer back to the instructions: The agent seemed to forget the
instructions to make the red paper. Always refer back to the instructions to
ensure you are following the correct steps.
```

Fig. 13: Meta-memory generated for ablation in GEN-ENV for a Chemistry task in Science-World.

| Task | Type | Generative L. Agents | | CLIN (ours) | |
|---|---|---|---|---|---|
| | | ReAct | Reflexion | BASE | ADAPT |
| Temp | S | 7.2 | 5.9 | 25.2 | 14.3 |
| Temp | S | 6.1 | 28.6 | 53.2 | 51.8 |
| Pick&Place | S | 26.7 | 64.9 | 92.5 | 100.0 |
| Pick&Place | S | 53.3 | 16.4 | 55.0 | 100.0 |
| Chemistry | S | 51.0 | 70.4 | 44.5 | 44.4 |
| Chemistry | S | 58.9 | 70.7 | 56.7 | 56.7 |
| Lifespan | S | 60.0 | 100.0 | 85.0 | 100.0 |
| Lifespan | S | 67.5 | 84.4 | 70.0 | 90.0 |
| Biology | S | 8.0 | 8.0 | 10.0 | 8.0 |
| Boil | L | 3.5 | 4.2 | 7.0 | 15.2 |
| Freeze | L | 7.8 | 7.8 | 10.0 | 10.0 |
| GrowPlant | L | 9.1 | 7.3 | 10.2 | 11.1 |
| GrowFruit | L | 18.6 | 13.0 | 35.9 | 71.6 |
| Biology | L | 27.7 | 2.6 | 70.0 | 81.0 |
| Force | L | 40.5 | 50.6 | 53.5 | 100.0 |
| Friction | L | 44.0 | 100.0 | 56.5 | 72.5 |
| Genetics | L | 25.7 | 50.9 | 77.4 | 100.0 |
| Genetics | L | 16.8 | 23.7 | 62.3 | 92.6 |
| | S | 37.6 | 49.9 | 54.7 | **62.8** |
| | L | 21.5 | 28.9 | 42.5 | **61.6** |
| | All | 29.6 | 39.4 | 48.6 | **62.2** |

Table 5: Comparing CLIN with baselines for **adaptation** in ScienceWorld

| Task | Type | RL Methods | | | Generative Language Agents | | | CLIN (ours) | | |
|---|---|---|---|---|---|---|---|---|---|---|
| | | DRRN | KGA2C | CALM | SayCan | ReAct | Reflexion | BASE | GEN-ENV | G+A |
| Temp | S | 6.6 | 6.0 | 1.0 | **26.4** | 7.2 | 5.9 | 25.2 | 15.7 | 13.8 |
| Temp | S | 5.5 | 11.0 | 1.0 | 8.0 | 6.1 | 28.6 | 53.2 | 49.7 | **58.2** |
| Pick&Place | S | 15.0 | 18.0 | 10.0 | 22.9 | 26.7 | 64.9 | 92.5 | 59.2 | 100.0 |
| Pick&Place | S | 21.7 | 16.0 | 10.0 | 20.9 | 53.3 | 16.4 | 55.0 | **100.0** | 100.0 |
| Chemistry | S | 15.8 | 17.0 | 3.0 | 47.8 | 51.0 | **70.4** | 44.5 | 42.2 | 51.7 |
| Chemistry | S | 26.7 | 19.0 | 6.0 | 39.3 | 58.9 | 70.7 | 56.7 | 85.6 | **93.3** |
| Lifespan | S | 50.0 | 43.0 | 6.0 | 80.0 | 60.0 | **100.0** | 85.0 | 65.0 | 100.0 |
| Lifespan | S | 50.0 | 32.0 | 10.0 | 67.5 | 67.5 | 84.4 | 70.0 | 75.0 | 90.0 |
| Biology | S | 8.0 | 10.0 | 0.0 | 16.0 | 8.0 | 8.0 | 10.0 | 32.0 | **32.0** |
| Boil | L | 3.5 | 0.0 | 0.0 | **33.1** | 3.5 | 4.2 | 7.0 | 4.4 | 16.3 |
| Freeze | L | 0.0 | 4.0 | 0.0 | 3.9 | 7.8 | 7.8 | **10.0** | 8.9 | 10.0 |
| GrowPlant | L | 8.0 | 6.0 | 2.0 | 9.9 | 9.1 | 7.3 | 10.2 | 10.9 | **11.2** |
| GrowFruit | L | 14.3 | 11.0 | 4.0 | 13.9 | 18.6 | 13.0 | 35.9 | 70.8 | **94.5** |
| Biology | L | 21.0 | 5.0 | 4.0 | 20.9 | 27.7 | 2.6 | 70.0 | 42.8 | **85.6** |
| Force | L | 10.0 | 4.0 | 0.0 | 21.9 | 40.5 | 50.6 | 53.5 | 70.0 | **100.0** |
| Friction | L | 10.0 | 4.0 | 3.0 | 32.3 | 44.0 | **100.0** | 56.5 | 70.0 | 94.0 |
| Genetics | L | 16.8 | 11.0 | 2.0 | 67.5 | 25.7 | 50.9 | 77.4 | 84.5 | **100.0** |
| Genetics | L | 17.0 | 11.0 | 2.0 | 59.5 | 16.8 | 23.7 | 62.3 | 61.4 | **100.0** |
| | **S** | 22.1 | 19.1 | 5.2 | 36.5 | 37.6 | 49.9 | 54.7 | 58.3 | **71.0** |
| | **L** | 11.2 | 6.2 | 1.9 | 29.2 | 21.5 | 28.9 | 42.5 | 47.1 | **68.0** |
| | **All** | 16.7 | 12.7 | 3.6 | 32.9 | 29.6 | 39.4 | 48.6 | 52.7 | **69.5** |

Table 6: Generalization across unseen environments in ScienceWorld

LLM (here, `gpt-4`). We find if we ablate for the controller module in CLIN, responsible for generating a goal before outputting the next action, CLIN's BASE performance drops in 44% cases. With an 18-point drop in average reward, the Abl-Contoller-BASE version of CLIN becomes equivalent to ReAct, the base agent for Reflexion, demonstrating the importance of the controller.

| Type | #trials to success ($\downarrow$) | %ep. improv. |
|---|---|---|
| S | 3.3 | 29.2 |
| L | 3.2 | 37.2 |
| All | 3.3 | 33.2 |

Table 7: CLIN's ADAPT improvements in ScienceWorld

| Type | GEN-TASK | | G + A | |
|---|---|---|---|---|
| | $\Delta$avg score | %ep. improv. | $\Delta$avg score | %ep. improv. |
| S | 14.6 | 40.0 | 4.9 | 5.7 |
| L | 10.3 | 36.7 | 9.2 | 15.6 |
| All | 13.0 | 38.8 | 6.5 | 9.3 |

Table 8: CLIN's GEN-TASK improvements in ScienceWorld

