# OpenReview forum: "CLIN: A Continually Learning Language Agent for Rapid Task Adaptation and Generalization"
_colmweb.org/COLM/2024/Conference — COLM_

### Official Review · Reviewer_8qBW · 2024-05-07

**Rating:** 7
**Confidence:** 4
**Ethics Flag:** 1

**Summary:**

This paper presents CLIN, a novel learning algorithm for LMM agents to reflect and improve on their collected experience by forming a memory of causal abstractions.  Expanding upon the _Reflexion_ algorithm, CLIN introduces dual memory updates to construct synergistic abstractions targeting two learning objectives, namely adaptation and generalization. Experimental validation conducted in two diverse textual environments, ALFWorld and Science World, underscores the efficacy of CLIN compared to Reflexion and other relevant baselines. The findings not only underscore the superiority of CLIN but also unveil insights into its underlying learning mechanisms.

Overall, this paper tackles an important problem within the field of decision-making systems and contributes to our comprehension of LLM's capabilities to form novel forms of reasoning. While a more detailed exploration of the role of language as an internal tool for decision-making systems could enhance the paper's positioning, it is well-motivated, clear, and supported by thorough experimentation.

**Questions To Authors:**

- I'm intrigued by the absence of algorithmic incentives for generating concise memories. LLMs often required constraints on their generation. Do you have an intuition on the underlying reasons for this phenomenon? The result section could investigate how often items are dropped in the memories.
- It is very informative to explain the kind of fuzzy matching and steps used to retrieve an action when the LLM does not provide a clear candidate. I’m curious as to how often this kind of fuzzy matching occurs during interactions.
- What are the shaded areas in figure 5?

**Reasons To Accept:**

The paper is very pedagogical and the results demonstrate the benefit of the proposed approach.

### Pedagogy

- The paper is, overall, very well-written. I enjoyed reading it.
- The paper does a great job of explaining the challenges of Scienceworld in terms of steps to achieve tasks, and convincingly justifies its relevance as a benchmark.
- Providing all the prompts used for the experiments is a strength. It should be the standard of every contribution leveraging LLM agents.

### Robust findings and thorough methodology

- Unlike previous approaches such as Reflexion, the authors also propose to emancipate from the gold trajectories provided by Scienceworld and use a zero-shot controller.
- The results section's organization in the form of questions offers clarity, allowing for a thorough assessment of CLIN's performance across diverse scenarios. It effectively isolates the impact of the two memory mechanisms and sheds light on optimal memory structures.

**Reasons To Reject:**

### Positioning

- Connecting the causal structure of memories with action models (Arorea et al 2018) is a relevant choice. However, the paper could include a brief discussion of previous approaches that use language as a means of abstraction for RL agents. This encompasses for instance approaches before LLM agents [1,2], as well as methodologies leveraging LLMs [3].
- The following perspective paper [4] could be pertinent in the discussion of applying linguistic abstraction to goal-directed agents (highlighted by authors on page 3).

### Method's limitation

- A minor limitation of the method is that, as always with the immense action space of text games, the approach (like ReAct and Reflexion) only works if we can provide the set of actions available at each time step.
- Experiments were only carried out using GPT-4 and could include other (maybe smaller) LLMs

### References

[1] Language as an Abstraction for Hierarchical Deep Reinforcement Learning, Yiding Jiang, Shixiang Gu, Kevin Murphy, Chelsea Finn, NeurIPS 2019

[2] Language as a Cognitive Tool to Imagine Goals in Curiosity-Driven Exploration. Cédric Colas, Tristan Karch, Nicolas Lair, Jean-Michel Dussoux, Clément Moulin-Frier, Peter Ford Dominey, Pierre-Yves Oudeyer, NeurIPS 2020

[3] Guiding Pretraining in Reinforcement Learning with Large Language Models, Yuqing Du, Olivia Watkins, Zihan Wang, Cédric Colas, Trevor Darrell, Pieter Abbeel, Abhishek Gupta, Jacob Andreas, ICML 2023

[4] Language and Culture Internalization for Human-Like Autotelic AI, Cédric Colas, Tristan Karch, Clément Moulin-Frier, Pierre-Yves Oudeyer, Nature Machine Intelligence

---

> ### Author Rebuttal · Authors · 2024-05-31
>
> Thank you for your review!
>
> `W1. Positioning: Connecting the causal structure of memories with action models (Arora et al. 2018) is a relevant choice. Could include a brief discussion of previous approaches that use language as a means of abstraction for RL agents...approaches before LLM agents [1,2], and methodologies leveraging LLMs [3]. This perspective paper [4] could be pertinent too.`
>
> This is a great point. We will include a discussion on the use of language as a form of abstraction in RL agents and cite the mentioned works; thank you for the pointers.
>
> `W2a. A minor limitation: With the immense action space of text games, the approach (like ReAct and Reflexion) only works if we can provide the set of actions available at each time step.`
>
> Again this is a good point, we will add a note in the paper.
>
> `W2b. Experiments were only carried out using GPT-4 and could include other (maybe smaller) LLMs`
>
> Thanks for pointing this out. We tried out with smaller LLMs but found they do not work at all in our setup, mostly because CLIN is a reasoning-based framework, which commonly requires a strong underlying LLM. Although we would like to have frameworks that work well with smaller LLMs, several previous reasoning frameworks also either require a strong proprietary model or find open-source LLMs less performant, similar to our findings.
>
> `Q1. I'm intrigued by the absence of algorithmic incentives for generating concise memories. LLMs often required constraints on their generation. An intuition on the underlying reasons for this phenomenon?`
>
> This is a great observation, and indeed was a surprise to us also: Indeed, the LM could simply keep growing the memory indefinitely, and we would need to externally apply constraints. We do not have a good explanation for this beyond the notion that LM's appear to have a "natural" level of detail behind them when providing an answer (e.g., GPT will answer a question with a paragraph, not an essay), and we are leveraging that to maintain a loosely constant-size memory. One interesting future work would be to encourage the memory to be larger or smaller, either through prompting or post-hoc filtering, and measure the effect of that.
>
> `Q2. How often fuzzy matching occurs`
>
> We observe the exact matching capability improves from GPT-3.5 to GPT-4.
>
> `Q3. What are the shaded areas in Figure 5?`
>
> The shaded region indicates the standard deviation in rewards or #steps across all environments (GEN-ENV) or tasks (GEN-TASK).

---

### Official Review · Reviewer_Vfjj · 2024-05-10

**Rating:** 5
**Confidence:** 4
**Ethics Flag:** 1

**Summary:**

The paper introduces CLIN, a memory-augmented, large language model (LLM)-based agent capable of continuous learning from interactions with simulation environments. The memory system records actions paired with their rewards to evaluate whether certain actions are necessary for achieving the final goals. Results indicate that this approach achieves state-of-the-art performance on benchmarks such as ScienceWorld and ALFWorld, surpassing existing agent task planning methods like Reflexion, React, and Saycan.

**Questions To Authors:**

The memory system depends on the agent's own exploration trails, so I wonder how the LLMs would affect the efficiency of the whole system? As I concern that if the agent cannot retrieve useful information or if the reward is consistently low for a bunch of trials, in the case that the agent cannot gain use knowledge of the world, what would happen.

**Reasons To Accept:**

1. The performance on benchmarks such as ScienceWorld and ALFWorld seems good, compared to other agent task planning methods like Reflexion, React, and Saycan.
2. The paper is well organized, the demonstration of the method is clear.

**Reasons To Reject:**

1. Misuse of the term continual learning: The term "continual learning" may be misleading. In machine learning, continual learning typically refers to a model's ability to learn from a sequence of different tasks, often requiring the model to behave as if not all contents were observed simultaneously. CLIN does not exhibit the characteristics of continual learning. The term "learning" as used here is brief and may be more accurately described as few-shot learning with trial-and-error iterations.
2. The assignment of rewards for given actions are not specified clearly. It is also good to study the influence using different feedback signals, reward scoring methods such as in Reflexion.
3. The authors should conduct experiments on more benchmark datasets such as hotpotQA, HumanEval.
4. How and where does the dynamic memory mechanism influence the model inference, and why it can help continue learning and adaptation to new tasks? More ablation study and case study should be given.

---

> ### Author Rebuttal · Authors · 2024-05-31
>
> Thank you for your review!
>
> `W1. "continual learning" misleading?`
>
> You are right. We are using "continual learning" in a slightly different sense, although the deviation isn't huge: In cross-episode learning (Sec 3.2), CLIN does learn from a sequence of __different__ tasks to perform on a new, previously unseen task, although they are all ScienceWorld related. To perform well on new tasks, CLIN does behave as if some contents are unobserved. It does this by selectively generalizing from prior episodes, effectively ignoring other prior knowledge.
>
> CLIN's memory mechanism is indeed quite different from continual parameter updates. However, the mechanism shouldn't be confused with the end result: CLIN's performance improves over changes in task and over time. In this sense, CLIN is an interesting alternative mechanism for performing a form of continual learning. Your comments are well-taken, and we will clarify these contrasts.
>
> `W2. How are rewards for given actions assigned?`
>
> Like many POMDP setups, CLIN only receives a reward at a trial's end. CLIN doesn't try to explicitly assign a numeric reward for individual actions; rather, the trials are stored, and when the memory is updated, CLIN relies on the LM to identify useful patterns in high-reward trials. In this way, the LM implicitly rewards some actions by generalizing them to memory.
>
> `W3. Experiment with hotpotQA, HumanEval?`
>
> For cross-task learning to make sense, there should be __some__ potential for learning in prior trials to help in new trials. For QA tasks like hotpotQA, it is not clear that this happens, as each question is very different.
>
> `W4a. How the memory influences inference`
>
> Memory affects model inference through prompting. By including retrievals in the prompt, CLIN essentially expands the instructions to the controller, providing additional information to help it make its next decision.
>
> `W4b. How dynamic memory helps learning/adaptation to new tasks`
>
> The memory helps adaptation by being continually updated after each trial, adding, modifying, and removing memories, ideally evolving to more helpful content (see ADAPT, Fig 5). Similarly, given a _new_ task, memory may help a little at the start (GEN, Fig 5), then as more trials are performed, it will again be updated, removing unhelpful content and evolving (G+A, Fig 5).
>
> `Q1. Memory depends on exploration?`
>
> Yes indeed. CLIN's negative memories (X does not contribute to Y) help steer it into new explorations when it gets stuck.

---

> > ### Author Response · Authors · 2024-06-07
> > **Following up.....**
> >
> > Just to follow up, we hope we have addressed your concerns, and that you might consider increasing your score. Please let us know if you still have unresolved questions, and thank you for your work serving as a reviewer!

---

### Official Review · Reviewer_d8nz · 2024-05-11

**Rating:** 6
**Confidence:** 4
**Ethics Flag:** 1

**Summary:**

The work proposes CLIN, which writes (create/edit) to a textual long-term memory semantic knowledge about the domain and its causal abstractions. Compared to existing work like Reflexion or Voyager, the key difference is that CLIN learns across environments and tasks.

On ScienceWorld and ALFWorld, it shows performance gain over the base agent and compared to baselines like ReAct and Reflexion, across setups of performing same/different task/environments.

**Questions To Authors:**

- Equation 1: what's e? I guess environment but not defined in 3.1.

- Table 1: Is SayCan basically "Act" in ReAct? i.e. "Observation" and "Action" turns? SayCan should have grounded visual parts, not sure if ScienceWorld has it.

**Reasons To Accept:**

- Long-term memory is a fundamental direction for language agents, and the paper makes solid contribution to this direction.

- The idea makes sense and is tested in systematic setups across domains, environments, and tasks.

- Findings are interesting, e.g. causal abstractions help more than free-form advice, the correlation of correctness and helpfulness of memories, etc. Some experiments and setups can serve as the basis for future research.

**Reasons To Reject:**

- The main limitation is that domains (especially ALFWorld) have regularities and are relatively synthetic. Thus, it is not surprising that causal abstractions can be learned and applied, given the current reasoning capabilities of LLMs. It would make the paper much stronger if some more noisy and realistic domains can be tested, e.g. Jericho/WebShop/WebArena/SWE-bench/etc. Even if the results are not good, I would still think the results make the paper stronger, as it objectively reflects limitations of current methods and better point out future directions.


- There are many math notations but the explanation is not most intuitive. I'd suggest some concrete examples to better illustrate the ideas.

---

> ### Author Rebuttal · Authors · 2024-05-31
>
> Thank you for your review!
>
> `W1. The main limitation is that domains (especially ALFWorld) have regularities and are relatively synthetic. ... It would make the paper much stronger if some more noisy and realistic domains can be tested, e.g. Jericho/WebShop/WebArena/SWE-bench/etc. Even if the results are not good, I would still think the results make the paper stronger...`
>
> Yes, ALFWorld is indeed very synthetic and regularized, with limited variation in both tasks and environments. However, Science World (where most of our effort lies) is at the other end of the spectrum, and is one of the more complex and varied text-based environments around at present, with tasks as varied as "grow and orange", "convert a liquid into a gas", etc.. Thus we hope we have at least covered two domain datapoints at these two extremes. The extensive experiments and ablations we performed on these two domains were at the limit of the scope of this work, both in terms of time and compute, and sufficient to demonstrate effectiveness and generality (at least across these different environments), but it would indeed be an interesting next step to explore generalization to other domains like WebArena.
>
> `W2. There are many math notations, I'd suggest some concrete examples to better illustrate the ideas`
>
> Thanks for the pointer. We will add this to the updated version.
>
> `Q1. Equation 1: what's e? I guess environment but not defined in 3.1`
>
> Yes, e is environment, as we define it in 3.2.
>
> `Q2. Table 1: Is SayCan basically "Act" in ReAct? i.e. "Observation" and "Action" turns? SayCan should have grounded visual parts, not sure if ScienceWorld has it.`
>
> Indeed, the SayCan baseline in ScienceWorld would be just an Act-equivalent of ReAct, because ScienceWorld does not provide visual information with observations.

---

> > ### Comment · Reviewer_d8nz · 2024-05-31
> >
> > Thanks for the rebuttal, i will keep my review as is

---

> > > ### Author Response · Authors · 2024-06-02
> > > **Thank you!**
> > >
> > > Dear reviewer,
> > >
> > > Thank you for your reply. Please let us know if you have any questions in the meantime.
> > >
> > > Thanks,\
> > > Authors

---

### Official Review · Reviewer_w3HD · 2024-05-13

**Rating:** 6
**Confidence:** 3
**Ethics Flag:** 1

**Summary:**

This paper proposes a general learning agent based on LLM to learn to act and adapt to within environments and across environments. The proposed agent is composed of memories, controller, executor, and memory generator. The four components build a pipeline for producing information necessary for interaction. The authors also adopts a special prompting template and claim causal abstraction is learned. The experiment results show a large boost over reflexion agents.

**Questions To Authors:**

see above.

**Reasons To Accept:**

+ The paper shows strong results over previous method (reflexion). The authors demonstrate the performance advantage across various setups.

+ The paper is clearly written with structured flows.

+ The proposed prompting template seems to be very useful

+ The idea of four components borrows the structure from RL community and makes sense.

**Reasons To Reject:**

+ The authors seem to have proposed a fairly complicated pipeline, making it difficult to understand which components actually contribute to the performance boost. It would be nice if the authors can thoroughly analyze, does the main boost come from new prompting template, executor design, etc.?

+ Following the above point, the contribution and takeaways are not clear for readers

+ Prompting agents are, interesting, useful, but a bit handwavy. I'm not sure how much of these tricks/methods can contribute in the long-term without sound statistical formulation for understanding and prompting LLMs.

---

> ### Author Rebuttal · Authors · 2024-05-31
>
> Thank you for your review!
>
> `W1. The pipeline is fairly complicated...Does the main boost come from new prompting template, executor design, etc.?`
>
> Despite having several components, CLIN is (we hope) somewhat of a minimalist extension on top of the standard language agent framework (Fig 2). We start with an executor similar to ReAct or Reflexion. We then found adding a separate controller that generates a goal for the next step (loosely analogous to ReAct’s “think” action) yields superior performance on ScienceWorld (Fig 4b) and similar performance on AlfWorld. We hypothesize that ReAct's few-shot demonstrations can often hurt performance; hence, our zero-shot base is a cleaner alternative where all performance comes from the cycles of goal generation and action execution.
>
> In addition to the baseline boost from adding a controller and working zero-shot, we observe memory (the remaining component) helps improve over time (Fig 4), and with a rate of improvement higher than RaAct and Reflexion. Finally, and perhaps most interestingly, the ablation in Section 5E suggests that structuring memory around causal abstractions (rather than freeform advice) has also helped performance.
>
> `W2. Clarify the contributions and takeaways`
>
> Yes, the two novel contributions are:
> (a) a continuously evolving memory (refined after each trial), rather than just a short-term retrieval from a prior trial (Reflexion), can help learning, including new tasks. We provide an architecture for implementing this.
> (b) Structuring memory around causal abstractions, rather than freeform advice (prior work), helps learning over an extended period
> We will better clarify these in the paper.
>
> `W3. Prompting agents are, interesting, useful, but a bit handwavy`
>
> Yes, indeed - the technology is a bit ad hoc and a sound statistical formulation for understanding prompting would be very helpful. However, our main goal is on the construction and use of non-parametric memory, with prompting as a mechanism for feeding those memories into new decisions. Despite the handwavyness, it appears to work well and achieves better generalization than finetuned models, making it interesting and significant for AI (e.g., a Behavior Cloning (Torabi et al., 2018) method presented in (Wang et al., 2022) was trained on 200K training examples to predict the next action. This method resulted in an average reward of 0.08 on the ScienceWorld benchmark). We hope the general prompting area achieves more rigor over time.

---

### Decision · Program_Chairs · 2024-07-10

**Decision:**

Accept

**Comment:**

This paper introduces CLIN, a language agent that leverages a memory of "causal abstractions" (rather than freeform "advice") to adapt to new tasks and environments. CLIN extends previous reflective language agents like ReAct or Reflexion by using a continuously evolving memory that is refined after each trial. The agent demonstrates strong performance gains over baselines on the ScienceWorld and ALFWorld benchmarks, both in adapting to repeated trials and in generalizing to new tasks and environments.

The reviewers generally agree that the paper tackles an important problem and makes a solid contribution to the developing area of language agents. The paper is well-written, the method is clearly described, and the experiments are thorough, including ablations that provide insights into the key components contributing to CLIN's performance. The proposed CLIN agent demonstrates strong empirical performance and the ability to learn within the same task/environment and adapt to new tasks/environments.

The authors have provided detailed and thoughtful responses to the reviewer comments in their rebuttal. They have offered to clarify the relationship of their approach to standard continual learning and to expand discussion of related work on linguistic abstractions in RL. While additional evaluations on more open-ended domains would be ideal, reviewers acknowledge that the experiments on ScienceWorld and ALFWorld already demonstrate effectiveness across rather different environment types.

Overall, the strengths of the paper - the novel dynamic memory architecture, strong empirical performance, and clear presentation - outweigh the limitations pointed out by the reviewers, and I tend to side with the more positive reviews. The paper makes a valuable contribution to the field and opens up promising directions for future work on language agents with more advanced "meta-memory" capabilities.

[At least one review was discounted during the decision process due to quality]